# Deep Wiener Deconvolution:
# Wiener Meets Deep Learning for Image Deblurring

**Jiangxin Dong**
MPI Informatics
jdong@mpi-inf.mpg.de

**Stefan Roth**
TU Darmstadt
stefan.roth@visinf.tu-darmstadt.de

**Bernt Schiele**
MPI Informatics
schiele@mpi-inf.mpg.de

## Abstract

We present a simple and effective approach for non-blind image deblurring, combining classical techniques and deep learning. In contrast to existing methods that deblur the image directly in the standard image space, we propose to perform an explicit deconvolution process in a feature space by integrating a classical Wiener deconvolution framework with learned deep features. A multi-scale feature refinement module then predicts the deblurred image from the deconvolved deep features, progressively recovering detail and small-scale structures. The proposed model is trained in an end-to-end manner and evaluated on scenarios with both simulated and real-world image blur. Our extensive experimental results show that the proposed *deep Wiener deconvolution network* facilitates deblurred results with visibly fewer artifacts. Moreover, our approach quantitatively outperforms state-of-the-art non-blind image deblurring methods by a wide margin.

## 1  Introduction

Image deblurring is a classical image restoration problem, which has attracted widespread attention [e.g., 1, 3, 9, 10, 56]. It is usually formulated as

$$y = x * k + n, \tag{1}$$

where $y, x, k$, and $n$ denote the blurry input image, the desired clear image, the blur kernel, and image noise, respectively. $*$ is the convolution operator. Traditional methods usually separate this problem into two phases, blur kernel estimation and image restoration (i.e., non-blind image deblurring). The goal of non-blind image deblurring is to restore the clear image $x$ from its corrupted observation $y$ given the blur kernel $k$. Early non-blind deblurring methods include the Wiener filter [49] and the Richardson-Lucy algorithm [33]. Later work commonly relied on a probabilistic formulation, most often maximum a-posteriori approaches, and much research has been devoted to developing effective image priors [15, 37, 50, 56] or sophisticated data terms [8, 32]. However, the optimization problems resulting from such advanced models are difficult to solve, which limits their practical appeal.

In recent years, convolutional neural networks have been exploited for image deblurring and shown promising results. To effectively exploit a-priori knowledge for non-blind image deblurring, one line of work [40, 42, 54, 55] decomposes this problem into image denoising (which is achieved by learned deep models) and image deconvolution[1] (performed in the standard image space). However, the underlying deep models for image denoising are not specifically optimized for image deblurring,

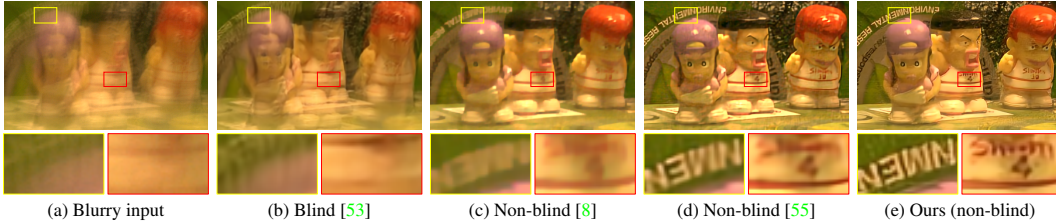

| (a) Blurry input | (b) Blind [53] | (c) Non-blind [8] | (d) Non-blind [55] | (e) Ours (non-blind) |

Figure 1: Deblurring results on a real blurry image from [50]. A recent blind image deblurring method [53], based on end-to-end trainable networks, does not effectively estimate a clear image. With an estimated blur kernel from [29], non-blind image deblurring methods [8, 55] generate better results *(c,d)* than the blind method *(b)*. Yet, our method recovers still much clearer results *(e)*.

thus not effective in removing deconvolution artifacts and restoring structural detail. In addition, we observe that performing the deconvolution in the standard image space introduces undesirable artifacts and loses fine-scale image detail (Fig. 1(d)). To remove artifacts, several algorithms use the information extracted by fixed feature extraction operators [31, 41] or discriminatively learned ones [8, 32] in the deconvolution process. However, these methods are not effective in finer-scale detail restoration (Fig. 1(c)). Another line of work adopts generic network architectures (e.g., U-Net [23, 45, 53], GANs [17]) to directly estimate the clear image from the blurry input and achieves reasonable image quality (without the blur kernel being known). Nevertheless, most of these networks do not perform well compared to established non-blind image deblurring methods if the blur kernel is known, cf. Fig. 1(b) *vs.* Fig. 1(c)–(d). Thus, it is of significant interest to investigate the properties of the deconvolution process and develop effective deep models for non-blind image deblurring.

In this paper, we develop a domain-specific network that integrates a classical deconvolution technique into deep neural networks for non-blind image deblurring. In this way, our approach deviates from existing methods [42, 55], whose deconvolution process is independent from the deep model. Specifically, we first explore the utility of the classical Wiener deconvolution and propose a *feature-based Wiener deconvolution*. Then we embed the feature-based Wiener deconvolution into a deep neural network, which contains a feature extraction network to provide useful features for the feature-based Wiener deconvolution and a feature refinement network to further refine the deconvolved features for better image reconstruction. Taken together, we find that the feature-based Wiener deconvolution can better constrain the whole network to learn effective non-blind deblurring.

The contributions of this paper can be summarized as follows: *(i)* We develop a novel feature-based Wiener deconvolution module that enforces the estimated latent image to coincide with the degradation process in a (deep) feature space. A detailed analysis demonstrates that a feature-space deconvolution is more effective in suppressing artifacts and recovering fine detail compared to previous methods that conduct the deconvolution in the image space. Learned deep features further improve the results. *(ii)* We then propose a multi-scale feature refinement module to restore the fine-scale structures of the deconvolved features, facilitating the reconstruction of high-quality images. The whole network is trained in an end-to-end manner. *(iii)* Benefitting from the feature-based Wiener deconvolution, our approach adaptively estimates the noise level from the blurry features, which ensures that training a single instance of the proposed *deep Wiener deconvolution network* is able to handle various levels of noise. Extensive experiments demonstrate that our approach outperforms existing state-of-the-art methods that require the noise level to be known by a large margin.

## 2  Related Work

**Non-learned methods.**  Since non-blind image deblurring is ill-posed, various priors have been proposed to constrain the solution space, e.g., total variation [47], hyper-Laplacian priors [15, 20], interal patch recurrence [25], etc. Danielyan et al. [5] develop an iterative deblurring algorithm based on BM3D [4]. Shan et al. [41] model the noise distribution by constraining several orders of its derivatives. Yuan et al. [52] propose an inter-scale and intra-scale deconvolution method to recover fine detail while suppressing artifacts. Common to these hand-crafted priors is that they do not fully exploit the characteristics of clean image data and usually lead to complicated inference problems.

**Classical learning methods.**  To better capture the inherent properties of clear photographic images, various learning methods have been proposed. Zoran and Weiss [56] present a Gaussian mixture

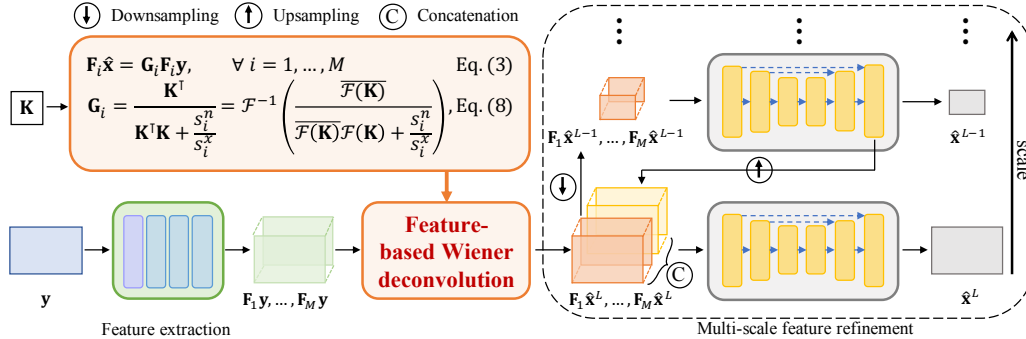

Figure 2: Deep Wiener deconvolution network. While previous work mostly relies on a deconvolution in the image space, our network first extracts useful feature information from the blurry input image and then conducts an explicit Wiener deconvolution in the (deep) feature space through Eqs. (3) and (8). A multi-scale encoder-decoder network progressively restores clear images, with fewer artifacts and finer detail. The whole network is trained in an end-to-end manner.

prior learned from natural images. Roth and Black [35] learn expressive high-order MRF priors, so called fields of experts. Schmidt et al. [39] derive a discriminative model based on regression tree fields [12]. Shrinkage fields [37] combine random fields with a discriminatively learned optimization algorithm. Similarly, Chen et al. [2] train nonlinear reaction diffusion models by parameterized linear filters and influence functions. To better model the image reconstruction error, [8, 32] learn a set of filters and penalty functions to model the data term. Although achieving decent image quality, these learned priors or penalty functions usually require the design of sophisticated numerical algorithms.

**Deep learning methods.** More recently, deep learning has been used for non-blind image deblurring. Xu et al. [51] use the singular value decomposition of the pseudo-inverse kernel to initialize the network parameters. However, this method needs fine-tuning for every kernel. To overcome this, several methods [34, 36, 40, 54, 55] decompose non-blind deblurring into two individual subproblems, i.e., image denoising and image deconvolution. For example, Zhang et al. [54] first deconvolve the blurry image in the image space and then use a fully convolutional network to learn image gradients to further guide the final deconvolution, while [40, 55] train a set of CNNs to improve the denoising accuracy. These methods achieve reasonable image quality, but separately designing these two subproblems will make the approach not be fully optimized for image deblurring. Kruse et al. [16] generalize discriminative FFT-based deconvolution by using CNN-based regularization and propose a boundary adjustment method. Gong et al. [10] incorporate deep neural networks into a gradient descent scheme. However, most of these approaches implement the deconvolution process in the standard image space. Hence, artifacts remain and fine-scale detail is lost, as pointed out by [8, 31, 41].

**Other related work.** Son and Lee [42] use a Wiener deconvolution as a preprocessing step and then develop residual networks with long and short skip-connections to remove artifacts caused by the Wiener deconvolution. Their network is able to remove artifacts but does not effectively preserve image detail. Thus, they adopt an additional postprocessing step to remedy this problem. Different from [42], we show that the Wiener deconvolution implemented in the standard image space is less effective for artifact suppression and detail restoration and thus propose a novel feature-based Wiener deconvolution embedded in an end-to-end architecture without any postprocessing step.

While most prior work assumes the noise level to be known, noise-blind methods offer an alternative. [38] develops a sampling-based Bayesian method with integrated noise estimation. [13] introduces a sampling-free approach using a smoothed generalization of the 0-1 loss. Different from [13, 38], we explore the Wiener deconvolution in a deep feature space and adaptively estimate the noise level from the blurry features, ensuring that training a single model is able to handle various levels of noise.

## 3 Deep Wiener Deconvolution Network

Our goal is to develop an effective non-blind image deblurring approach to restore high-quality images with few artifacts and fine detail. Specifically, the proposed deep Wiener deconvolution

network contains a feature-based Wiener deconvolution module and a multi-scale feature refinement module, which are trained in an end-to-end manner. Figure 2 summarizes the model architecture.

## 3.1 Feature-based Wiener deconvolution

To design a network specifically for the task of image deblurring, we propose to embed an explicit Wiener deconvolution step into our network. However, only using the intensity information in the standard image space for the deconvolution is not effective for artifact removal and detail restoration as pointed out by [8, 31, 41]. Thus, we propose a *feature-based Wiener deconvolution module* to better constrain the deconvolution process with useful feature information from the blurry input.

Let $\{f_i\}_{i=1}^M$ denote a set of linear filters, which are used to extract useful feature information from the blurry input. By convolving both sides of Eq. (1) with $\{f_i\}$, we can obtain the relationship between $y, x$, and $k$ in the feature space, owing to the properties of convolution,

$$\mathbf{F}_i\mathbf{y} = \mathbf{K}\mathbf{F}_i\mathbf{x} + \mathbf{F}_i\mathbf{n}, \quad \forall i = 1, \dots, M, \tag{2}$$

where $\mathbf{F}_i, \mathbf{K}, \mathbf{y}, \mathbf{x}$, and $\mathbf{n}$ denote the matrix/vector forms of $f_i, k, y, x$, and $n$.

The goal of our feature-based Wiener deconvolution module is to explicitly deconvolve the blurry features $\{\mathbf{F}_i\mathbf{y}\}$ from Eq. (2) by finding a set of *feature-based Wiener deconvolution operators* $\{\mathbf{G}_i\}$ so that we can obtain the latent features as

$$\mathbf{F}_i\hat{\mathbf{x}} = \mathbf{G}_i\mathbf{F}_i\mathbf{y}, \quad \forall i = 1, \dots, M, \tag{3}$$

where $\hat{\mathbf{x}}$ is the latent clear image sought after. In order to obtain latent features that are close to the ground-truth clear features, for each $i$, we need to minimize the mean squared error [49]

$$e_i = \mathbb{E}\big(|\mathbf{F}_i\mathbf{x} - \mathbf{F}_i\hat{\mathbf{x}}|^2\big) = \mathbb{E}\big(|\mathbf{F}_i\mathbf{x} - \mathbf{G}_i\mathbf{F}_i\mathbf{y}|^2\big) = \mathbb{E}\big(|\mathbf{F}_i\mathbf{x} - \mathbf{G}_i(\mathbf{K}\mathbf{F}_i\mathbf{x} + \mathbf{F}_i\mathbf{n})|^2\big) \tag{4a}$$

$$= (1 - \mathbf{G}_i\mathbf{K})(1 - \mathbf{G}_i\mathbf{K})^\top \mathbb{E}\big(|\mathbf{F}_i\mathbf{x}|^2\big) - (1 - \mathbf{G}_i\mathbf{K})\mathbf{G}_i^\top \mathbb{E}\big(\mathbf{F}_i\mathbf{x}(\mathbf{F}_i\mathbf{n})^\top\big) \tag{4b}$$

$$- (1 - \mathbf{G}_i\mathbf{K})^\top \mathbf{G}_i\mathbb{E}\big((\mathbf{F}_i\mathbf{x})^\top \mathbf{F}_i\mathbf{n}\big) + \mathbf{G}_i\mathbf{G}_i^\top \mathbb{E}\big(|\mathbf{F}_i\mathbf{n}|^2\big),$$

where $\mathbb{E}$ denotes the expectation. Assuming the noise to be independent from the latent clear image and having zero mean, we can derive that

$$\mathbb{E}\big(\mathbf{F}_i\mathbf{x}(\mathbf{F}_i\mathbf{n})^\top\big) = \mathbb{E}\big(\mathbf{F}_i\mathbf{x}\big)\mathbb{E}\big((\mathbf{F}_i\mathbf{n})^\top\big) = 0 \quad \text{and} \quad \mathbb{E}\big((\mathbf{F}_i\mathbf{x})^\top \mathbf{F}_i\mathbf{n}\big) = \mathbb{E}\big((\mathbf{F}_i\mathbf{x})^\top\big)\mathbb{E}\big(\mathbf{F}_i\mathbf{n}\big) = 0. \tag{5}$$

We denote $\mathbb{E}\big(|\mathbf{F}_i\mathbf{x}|^2\big)$ and $\mathbb{E}\big(|\mathbf{F}_i\mathbf{n}|^2\big)$ by $s_i^x$ and $s_i^n$ and then rewrite Eq. (4b) as

$$e_i = \big(1 - \mathbf{G}_i\mathbf{K}\big)\big(1 - \mathbf{G}_i\mathbf{K}\big)^\top s_i^x + \mathbf{G}_i\mathbf{G}_i^\top s_i^n. \tag{6}$$

To minimize $e_i$, we compute the derivative of Eq. (6) with respect to $\mathbf{G}_i$ and set it to zero:

$$\big(\mathbf{K}^\top\mathbf{K}s_i^x + s_i^n\big)\mathbf{G}_i - \mathbf{K}^\top s_i^x = 0. \tag{7}$$

Then we can obtain the feature-based Wiener deconvolution operator $\mathbf{G}_i$ as

$$\mathbf{G}_i = \frac{\mathbf{K}^\top}{\mathbf{K}^\top\mathbf{K} + \frac{s_i^n}{s_i^x}} = \mathcal{F}^{-1}\left(\frac{\overline{\mathcal{F}(\mathbf{K})}}{\overline{\mathcal{F}(\mathbf{K})}\mathcal{F}(\mathbf{K}) + \frac{s_i^n}{s_i^x}}\right), \tag{8}$$

where $\mathcal{F}$ denotes the discrete Fourier transform and $\overline{\mathcal{F}(\mathbf{K})}$ is the complex conjugate of $\mathcal{F}(\mathbf{K})$. Thus, we can obtain the latent feature $\{\mathbf{F}_i\hat{\mathbf{x}}\}$ by Eqs. (3) and (8).

To extract useful feature information from the blurry input, we can choose common derivative operators (e.g., first- and higher-order derivatives) or discriminatively learned linear filters. In general, the input of the feature-based Wiener deconvolution module is the degraded image and the outputs are the deconvolved latent features $\{\mathbf{F}_i\hat{\mathbf{x}}\}$. Our analysis in Sec. 5 shows that the Wiener deconvolution is more effective when combining more and more useful feature information.

In addition, we can obtain more powerful feature extractors $\{\mathbf{F}_i\}$ in Eq. (3) using deep neural networks [6, 26, 28]. While deep feature extractors are not linear, as assumed by Eq. (2), they are locally linear [19, 27]. Hence, we apply Eq. (2) regardless; remaining errors can be compensated by the feature refinement (Sec. 3.2). We directly estimate the blurry features $\{\mathbf{F}_i\mathbf{y}\}$ given as input to the deconvolution $\mathbf{G}_i$ and use a feature extraction network with one convolutional layer followed by

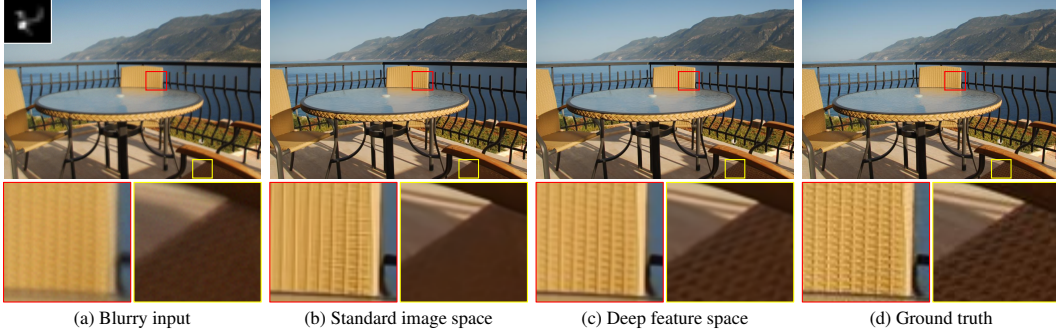

| (a) Blurry input | (b) Standard image space | (c) Deep feature space | (d) Ground truth |

Figure 3: Effect of the feature-based Wiener deconvolution. *(a)* Blurry image and blur kernel. *(b)* and *(c)* are the results by methods that perform the deconvolution in the standard image space and a deep feature space, respectively. *(d)* Ground truth. The deblurred result *(c)* from the proposed approach contains fewer artifacts and much more detail in the red and yellow boxes than those in *(b)*.

three residual blocks [11]. By embedding the feature-based Wiener deconvolution into an end-to-end network, the feature extraction network can learn more useful features for deconvolution, cf. Sec. 5.

We estimate $s_i^x$ as the standard deviation of the blurry feature $\mathbf{F}_i \mathbf{y}$ and estimate $s_i^n$ as the variance of the difference between the blurry feature $\mathbf{F}_i \mathbf{y}$ and the mean-filtered result of $\mathbf{F}_i \mathbf{y}$. A more detailed analysis is included in the supplemental material. $s_i^n$ can approximately estimate the noise level of each blurry feature. Benefitting from the feature-based Wiener deconvolution and the multi-scale feature refinement in Sec. 3.2, our network is able to handle blurry images with various noise levels.

To intuitively illustrate the effect of the feature-based Wiener deconvolution module, we compare performing the Wiener deconvolution in the standard image space and in a deep feature space in Fig. 3(b) and (c). For fair comparison, both methods use the same multi-scale feature refinement in Sec. 3.2 to reconstruct the final results. Figure 3 shows that deconvolving the blurry image in a deep feature space is much more effective at yielding a clear image with fewer artifacts (e.g., the chair back in the red boxes of Fig. 3) and finer textures (e.g., the seat in the yellow boxes of Fig. 3).

### 3.2 Multi-scale feature refinement

To restore high-quality images from the deconvolved latent features, we develop a multi-scale feature refinement module in a coarse-to-fine manner. Specifically, we first build a pyramid $\{\{\mathbf{F}_i \hat{\mathbf{x}}^l\}_{l=1}^L\}$ of the full-resolution latent features $\{\mathbf{F}_i \hat{\mathbf{x}}\}$ from Eq. (3) using bicubic downsampling with a scale factor of 2. Then we can obtain the clear image $\hat{\mathbf{x}}^l$ at each scale by

$$\hat{\mathbf{x}}^l = \mathcal{N}(\mathbf{h}^l), \quad \mathbf{h}^l = \begin{cases} \mathcal{C}\big(\mathbf{F}_1 \hat{\mathbf{x}}^l, \ldots, \mathbf{F}_M \hat{\mathbf{x}}^l\big) & \text{if } l = 1, \\ \mathcal{C}\big(\mathbf{F}_1 \hat{\mathbf{x}}^l, \ldots, \mathbf{F}_M \hat{\mathbf{x}}^l, \mathcal{N}_{-1}(\mathbf{F}_1 \hat{\mathbf{x}}^{l-1}, \ldots, \mathbf{F}_M \hat{\mathbf{x}}^{l-1}) \uparrow \big) & \text{if } l = 2, \ldots, L, \end{cases} \quad (9)$$

where $\mathcal{N}$ denotes a network for the feature refinement module, $\mathcal{N}_{-1}$ denotes the network $\mathcal{N}$ without the last layer, $\mathcal{C}$ is the concatenation, and $\uparrow$ is the upsampling operation. For the network $\mathcal{N}$, we use an encoder-decoder architecture to refine the latent features and reconstruct the final clear image. The detailed network parameters are given in the supplemental material.

**Loss function.** To better regularize the network, we apply an $\ell_1$ loss function at each scale $l$. The final loss function is given as

$$\mathcal{L} = \sum_{l=1}^{L} \frac{\gamma_l}{N_l} \big\| \hat{\mathbf{x}}^l - \mathbf{x}^l \big\|_1, \quad (10)$$

where $\mathbf{x}^l$ is the downsampled ground-truth image using bicubic interpolation for the scale $l$, $\{\gamma_l\}$ are the weights for each scale, and $N_l$ is the number of elements in $\mathbf{x}^l$ for normalization.

Table 1: Quantitative comparison to state-of-the-art methods on the dataset of Levin et al. [21].

| Noise level | | DMPHN [53] | EPLL [56] | MLP [40] | CSF [37] | LDT [8] | FCN [54] | IRCNN [55] | FDN [16] | FNBD [42] | RGDN [10] | Ours |
|---|---|---|---|---|---|---|---|---|---|---|---|---|
| 1% | PSNR (dB) | 25.95 | 34.06 | 32.08 | 31.90 | 31.53 | 33.22 | 34.33 | 34.05 | 34.81 | 33.96 | **36.90** |
| | SSIM | 0.7918 | 0.9310 | 0.8884 | 0.9024 | 0.8977 | 0.9267 | 0.9210 | 0.9335 | 0.9398 | 0.9395 | **0.9614** |
| 3% | PSNR (dB) | 25.78 | 29.09 | 27.00 | 28.01 | 28.39 | 29.49 | 30.04 | 29.77 | 30.63 | 29.71 | **32.77** |
| | SSIM | 0.7814 | 0.8460 | 0.7016 | 0.8013 | 0.8052 | 0.8599 | 0.8156 | 0.8583 | 0.8658 | 0.8662 | **0.9179** |
| 5% | PSNR (dB) | 25.19 | 26.54 | 25.38 | 26.32 | 26.70 | 27.72 | 28.51 | 27.94 | 27.93 | 27.45 | **30.77** |
| | SSIM | 0.7579 | 0.7785 | 0.6330 | 0.7427 | 0.7468 | 0.8142 | 0.7762 | 0.8139 | 0.7759 | 0.7889 | **0.8857** |
| 10% | PSNR (dB) | 22.68 | 23.47 | 23.86 | 23.18 | 23.87 | 24.46 | 26.14 | – | 23.98 | 21.54 | **27.78** |
| | SSIM | 0.6269 | 0.6639 | 0.5799 | 0.5820 | 0.5919 | 0.6859 | 0.7078 | – | 0.5756 | 0.4642 | **0.8189** |

Table 2: Quantitative comparison to state-of-the-art methods on the datasets of [43] and [24].

| Dataset | | DMPHN [53] | EPLL [56] | MLP [40] | CSF [37] | LDT [8] | FCN [54] | IRCNN [55] | FDN [16] | FNBD [42] | RGDN [10] | Ours |
|---|---|---|---|---|---|---|---|---|---|---|---|---|
| [43] | PSNR (dB) | 24.75 | 32.48 | 31.47 | 31.52 | 30.52 | 32.36 | 33.57 | 32.63 | 31.22 | 31.25 | **34.05** |
| | SSIM | 0.7111 | 0.8815 | 0.8535 | 0.8622 | 0.8399 | 0.8853 | 0.8977 | 0.8887 | 0.8860 | 0.8869 | **0.9225** |
| [24] | PSNR (dB) | 24.21 | 29.81 | 28.47 | 29.00 | 28.20 | 29.51 | 30.63 | 29.93 | 30.92 | 29.51 | **31.74** |
| | SSIM | 0.6572 | 0.8385 | 0.7977 | 0.8230 | 0.7922 | 0.8339 | 0.8645 | 0.8555 | 0.8799 | 0.8616 | **0.8938** |

## 4 Experimental Results

Next, we first describe the datasets and implementation details of our deep Wiener deconvolution network. Then, we evaluate the proposed method against the state of the art on images with both simulated and real-world blur. More experimental results are included in the supplemental material.

### 4.1 Datasets and implementation

**Training dataset.** We collect a training dataset including $400$ images from the Berkeley segmentation [24] and $4744$ images from the Waterloo Exploration [22] datasets. Specifically, we generate the training dataset as is common in the non-blind deblurring literature [54, 55]. We randomly crop patches of $256 \times 256$ pixels from each clear image. Then we synthesize realistic kernels [39] of random sizes in the range from $13 \times 13$ to $35 \times 35$ pixels and convolve each clear image patch with a blur kernel. Finally, we add Gaussian noise to the blurry images with noise levels ranging from $0$ to $5\%$. We train a single model for *all noise levels* and *without fine-tuning* for each test scenario.

**Test datasets.** We first use the popular benchmark test datasets of Levin et al. [21] and Sun et al. [43] to evaluate our approach. These two datasets contain 4 and 80 clear images, respectively, while adopting the same 8 blur kernels from [21]. Next, the proposed method is evaluated on a test dataset generated using 100 clear images from the dataset of [24] and 100 synthetic blur kernels from [39]. On the dataset of Levin et al. [21], we evaluate all methods with different Gaussian noise levels ($1\%, 3\%, 5\%,$ and $10\%$). For the test datasets of [43] and [24], we add $1\%$ Gaussian noise. The training dataset and all test datasets do not overlap.

**Implementation details.** Balancing effectiveness and efficiency, we use a total of two scales in the multi-scale feature refinement module. We empirically use $M = 16$ features and set $\gamma_l = 1$. For training the network parameters, we adopt the Adam optimizer [14] with default parameters. The batch size is set to 8. The learning rate is initialized as $10^{-4}$, which is halved every 200 epochs. The PyTorch code and trained models are available at our Project page. As pointed out in [29, 44], the non-linear camera response function can be addressed before the deconvolution steps, hence we do not consider these factors in our experiments. Although our method is derived based on a uniform blur model, it can be extended to handle non-uniform blur by applying the local uniform approximation method [48]. More discussions and experimental results for non-uniform blur are included in the supplemental material.

### 4.2 Results with simulated blur

We compare our approach against several state-of-the-art methods including DMPHN [53], EPLL [56], MLP [40], CSF [37], LDT [8], FCN [54], IRCNN [55], FDN [16], FNBD [42], and RGDN [10]. For fair comparison, we retrain the methods [8, 37, 53] with the same training dataset as ours and use the implementation of other methods available online, tuning their adjustable parameters for different datasets and noise levels for best possible results.

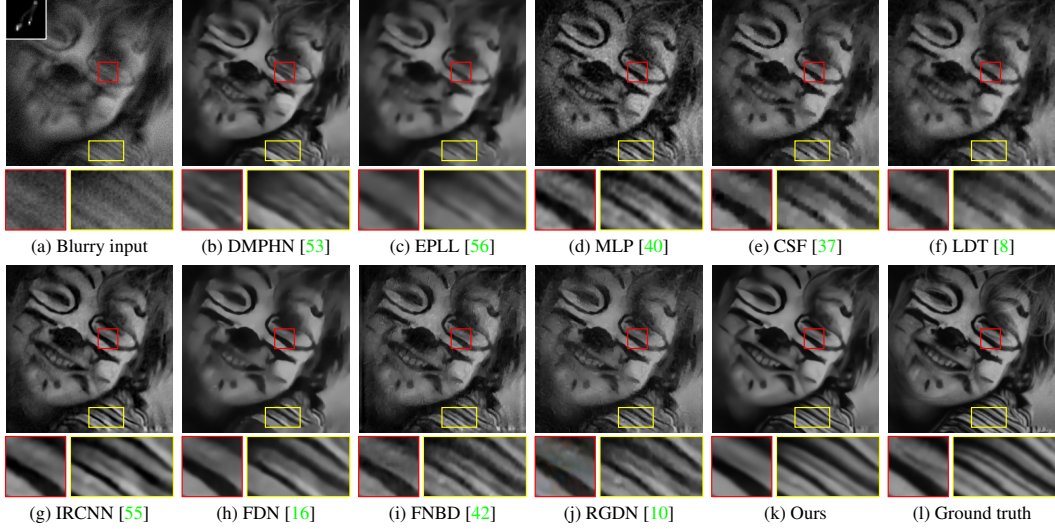

(a) Blurry input    (b) DMPHN [53]    (c) EPLL [56]    (d) MLP [40]    (e) CSF [37]    (f) LDT [8]

(g) IRCNN [55]    (h) FDN [16]    (i) FNBD [42]    (j) RGDN [10]    (k) Ours    (l) Ground truth

Figure 4: Example with simulated blur (5% noise level) from the dataset of [21]. The result obtained by [40] has severe artifacts in *(d)*. For other methods, small-scale structures and detail are over-smoothed as shown in the red and yellow boxes of *(c)–(j)*. Compared to existing methods, our approach can effectively preserve finer detail as shown in *(k)*.

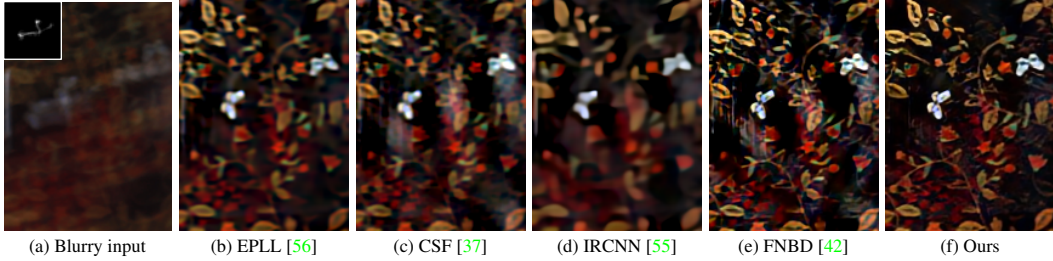

(a) Blurry input    (b) EPLL [56]    (c) CSF [37]    (d) IRCNN [55]    (e) FNBD [42]    (f) Ours

Figure 5: Example with real camera shake from [29]. Compared with the results in *(b)–(e)*, the deblurred image *(f)* generated by our approach is much clearer with finer detail and fewer artifacts.

We first examine our approach on the dataset of [21] in Tab. 1. Without manual parameter tuning, our method outperforms competing approaches by a wide margin, increasing the PSNR by at least 1.64dB across the noise levels. FNBD [42] achieves favorable results among the competing methods. It uses a Wiener deconvolution as preprocessing, a neural network to remove artifacts, and a postprocessing step to recover detail. In contrast, we explore the Wiener deconvolution in a deep feature space and embed it into our end-to-end network, which is clearly more effective. Note that while our training dataset only contains blurry images with Gaussian noise of 0–5%, our method still performs better for blurry images with 10% Gaussian noise than competing methods, highlighting its robustness.

Figure 4 shows a visual comparison. A state-of-the-art blind deblurring method [53] based on an end-to-end network cannot recover a clear image (Fig. 4(b)), showing the need to further research non-blind methods and backbones. The results of existing non-blind methods, however, also do not fully recover fine-scale structures or contain severe artifacts (Fig. 4(c)-(j)). In contrast, our deep Wiener deconvolution yields a much clearer image with finer detail and fewer artifacts (Fig. 4(k)). As discussed in Sec. 5, our improved results mainly stem from the domain-specific network for image deblurring, which explicitly embeds a feature-based Wiener deconvolution into a deep network.

We then evaluate the proposed approach on the datasets of [43] and [24] in Tab. 2 and again observe a significant gain. The average PSNR of our method is at least 0.82dB higher than the best result of [42] on the dataset of [24] and at least 0.48dB higher than the best competing method on the dataset by Sun et al. [43]. More visual comparisons are included in the supplemental material.

Table 3: Effectiveness of the feature-based Wiener deconvolution. All methods are evaluated on the dataset of [21] with $1\%$ Gaussian noise, where the kernel size ranges from $13 \times 13$ to $27 \times 27$ pixels. The basic reconstruction network that contains three residual blocks followed by one convolutional layer is denoted as *Basic reconstruction*.

| | Feature extraction | | | Wiener deconvolution | Refinement / Reconstruction | | PSNR (dB) /SSIM |
|---|---|---|---|---|---|---|---|
| | Intensity | Gradients | Deep features | | Basic reconstruction | Multi-scale refinement | |
| Wiener [49] | ✔ | ✗ | ✗ | ✔ | ✗ | ✗ | 27.48/0.6981 |
| Wiener$_D$ | ✗ | ✗ | ✔ | ✔ | ✔ | ✗ | 34.78/0.9478 |
| Ours$_{\text{w/o Wiener}}$ | ✗ | ✗ | ✔ | ✗ | ✗ | ✔ | 25.70/0.7893 |
| Ours$_I$ | ✔ | ✗ | ✗ | ✔ | ✗ | ✔ | 35.13/0.9438 |
| Ours$_G$ | ✗ | ✔ | ✗ | ✔ | ✗ | ✔ | 28.79/0.9031 |
| Ours$_{I+G}$ | ✔ | ✔ | ✗ | ✔ | ✗ | ✔ | 36.52/0.9583 |
| Ours$_{9\text{ features}}$ | ✗ | ✗ | ✔ | ✔ | ✗ | ✔ | 36.81/0.9608 |
| Ours | ✗ | ✗ | ✔ | ✔ | ✗ | ✔ | **36.90/0.9614** |

## 4.3 Results with real blur

We further evaluate our method on images with real camera shake. One example from [29] is shown in Fig. 5, where the blur kernel is estimated by [7]. The results by [37, 42] exibit visible artifacts, cf. Fig. 5(c) and (e). The methods [55, 56] over-smooth the detail in the deblurred images, see Fig. 5(b) and (d). In contrast, our deep Wiener deconvolution network outputs a much clearer image, cf. Fig. 5(f), where the fine-scale structures of the butterflies and leaves are recovered much better. More comparisons are included in the supplemental material.

## 5 Analysis and Discussion

**Effectiveness of the feature-based Wiener deconvolution.** To demonstrate the effect of the deep feature information on the deconvolution with the Wiener framework, we first compare the classical Wiener deconvolution [49] with a baseline method that combines [49] with learned deep features (*Wiener$_D$* for short). To reconstruct the final image from the deconvolved deep features for this baseline method, we use a network of three residual blocks followed by one convolutional layer (i.e. the inverse of the feature extraction network, *Basic reconstruction* for short) to go back to the image space. We then use the dataset [21] ($1\%$ noise level) as described in Sec. 4.1 for evaluation. Table 3 shows the quantitative comparison, where we find that the Wiener deconvolution is more effective when combined with deep features, cf. $27.48$dB for [49] *vs.* $34.85$dB for *Wiener$_D$*. Also note that our encoder-decoder based multi-scale refinement network is much better at recovering the clear image from the deconvolved features, increasing the PSNR by $2.05$dB to $36.90$dB.

To further analyze the effect of the proposed feature-based Wiener deconvolution module, we individually remove the Wiener deconvolution (*Ours$_{\text{w/o Wiener}}$* for short, which contains the feature extraction network followed by the feature refinement network and thus is not guided by the blur kernel) or the feature extraction (*Ours$_I$* for short) and train these baseline models using the same settings as ours. Table 3 shows that the PSNR of our full network is at least $1.77$dB higher than each baseline method. The comparison results illustrate the importance of the feature-based Wiener deconvolution on the end-to-end network for non-blind image deblurring, which can effectively incorporate kernel information and leverage deep features.

We additionally investigate whether using the commonly used gradient information [31, 41] can already generate competitive results with a Wiener deconvolution. To that end, we further compare with baseline methods that respectively replace the learned deep features with the image gradients along the vertical and horizontal directions (*Ours$_G$* for short) and the concatenation of both the blurry image and the image gradients (*Ours$_{I+G}$* for short) in our implementation. The results in Tab. 3 demonstrate that gradient information alone is not sufficient, and only yields good results when combined with intensity information in the standard image space. Yet, our learned deep features outperform this manual feature combination in the feature-based Wiener deconvolution by about $0.4$dB. To further ensure a fair comparison, we also train a model that learns 9 features in the feature extraction network (*Ours$_{9\text{ features}}$* for short), since the combination of the raw image and the image gradients yields 9 feature channels. As Tab. 3 shows, both the proposed networks with 9 and 16 features perform substantially better than all other baseline methods, which demonstrates

Table 4: Effect of the multi-scale feature refinement (PSNR in dB/SSIM). All methods are evaluated on the dataset of [21] with $1\%$ Gaussian noise, where the kernel size ranges from $13 \times 13$ to $27 \times 27$ pixels. We denote the baseline method without the multi-scale refinement as *Ours$_{w/o\ multi\text{-}scale}$*.

| Noise level | Kernel size | 13–19 | 33–39 | 53–59 |
|---|---|---|---|---|
| 1% | Ours$_{w/o\ multi\text{-}scale}$ | 33.46/0.9232 | 29.23/0.8398 | 27.45/0.7995 |
|  | Ours | 33.57/0.9254 | 29.47/0.8437 | 27.73/0.8127 |
| 3% | Ours$_{w/o\ multi\text{-}scale}$ | 29.93/0.8475 | 26.71/0.7370 | 25.20/0.6918 |
|  | Ours | 30.09/0.8519 | 27.00/0.7510 | 25.57/0.7080 |

Table 5: Robustness to inaccurate blur kernels (PSNR in dB/SSIM). All the methods are evaluated on the dataset of [18] with $1\%$ Gaussian noise, where the kernel size ranges from $51 \times 51$ to $101 \times 101$ pixels.

| Methods | Kernels for training | Kernels for testing | |
|---|---|---|---|
|  |  | GT | [50] |
| Vasu et al. [46] | – | – | 23.00/0.7554 |
| Ours | GT | 27.75/0.8664 | 23.74/0.7998 |
| Ours | [46] | 25.55/0.8298 | 23.45/0.7834 |
| Ours | Mix of GT & [46] | 27.33/0.8620 | 23.80/0.7987 |

that the feature extraction network can effectively learn useful features for the embedded Wiener deconvolution step to facilitate better image deblurring.

Compared to the classical image-based Wiener deconvolution, the feature-based one can utilize more useful information beyond the image intensity to better constrain the deconvolution process. In addition, finer-scale detail is better modeled in the feature space. Particularly, the deep feature extractor in our end-to-end network can adaptively learn useful features for high-quality image deblurring (Fig. 3). More analyses are included in the supplemental material.

**Effectiveness of the multi-scale feature refinement.** Although multi-scale approaches have been widely used in blind image deblurring [30, 45] to facilitate extracting multi-scale structural information, its effect on non-blind deblurring based on deep learning is still unknown. To further demonstrate the effect of the multi-scale refinement, we compare with a baseline method without using the multi-scale refinement (i.e. we set $L = 1$ in Eqs. (9) and (10), *Ours$_{w/o\ multi\text{-}scale}$* for short) that is implemented in the same way as ours otherwise, and evaluate on the dataset of [24] with different kernel sizes and noise levels. Table 4 shows that, depending on the kernel size in the ranges 13–19, 33–39, and 53–59, the PSNR of our method is 0.16dB, 0.29dB, and 0.37dB higher, respectively (with $3\%$ Gaussian noise), than the baseline method without using multi-scale refinement. This demonstrates that our multi-scale refinement clearly improves the results, especially in the challenging cases of large blurs (in which many methods including blind ones are known to have difficulties) and strong noise.

**Robustness to inaccurate kernels.** Blur kernels estimated by blind image deblurring methods usually contain substantial errors. Thus, it is necessary to evaluate the robustness of our proposed non-blind method to inaccurate kernels. We use the same method as described in Sec. 4.1 to generate the training data. To train our model for this task, we respectively use GT kernels, synthetic noisy kernels by [46], and a mix of GT and synthetic noisy kernels. We then compare with the method of [46], which focuses on handling inaccurate kernels and use the same test dataset [18] as [46] for evaluation. The dataset [18] contains $1\%$ Gaussian noise and the kernel size ranges from $51 \times 51$ to $101 \times 101$ pixels. Table 5 shows that the proposed approach is robust to inaccurate kernels, even when trained with GT kernels, improving the results over [46] by a large margin of 0.7dB.

Compared to competing approaches, our method performs robustly to noise and inaccurate kernels for two reasons. First, benefitting from the proposed feature-based Wiener deconvolution, our method can adaptively estimate the noise-related parameters $s_i^x$ and $s_i^n$ as stated in Sec. 3.1. Second, our end-to-end network facilitates the feature extractor for learning useful features for deconvolution with fewer artifacts, as well as benefits from the feature refinement module that uses redundant deconvolved features to reconstruct clearer images. The robustness is also evaluated on real-world images as shown in Figs. 1 and 5. More comparisons are included in the supplemental material.

# 6 Conclusion

In this paper, we propose a feature-based Wiener deconvolution module, where we explore useful learned features from deep neural networks to yield an explicit Wiener deconvolution step in the extracted deep feature space. We show that compared with the commonly used deconvolution conducted in the image space, it is much more effective to deconvolve the blurry image in a (deep) feature space. We further develop a multi-scale feature refinement module to progressively restore fine-scale detail from the deconvolved features. Extensive evaluations and comparison with state-of-the-art methods demonstrate that our approach extends the state of the art in non-blind deblurring by a wide margin while being robust to different noise levels and inaccurate blur kernels.

## Broader Impact

Since blur is a common artifact in imaging systems, such as from the point spread function of the optical system, image deblurring has a *broad potential impact* through a wide range of applications. These include *satellite imaging, medical imaging, telescope imaging in astronomy, and portable device imaging*. Our image deblurring technique based on the proposed deep Wiener deconvolution network can provide high-quality clear images to facilitate intelligent data analysis tasks in these fields and it is apparent that applications, e.g., in medical imaging or portable device imaging have significant societal impact. To illustrate its applicability, we provide some examples for potential applications of our approach in the supplemental material.

Despite the many benefits of high-quality image deblurring, negative consequences can still arise, largely because image deblurring can present certain *risks to privacy*. For example, in order to protect the privacy of certain individuals depicted in visual media, such as on TV or in the press, their depiction will sometimes be blurred artificially to hide the individual's identity. In this case, deblurring can pose the risk of unhiding the person's identity, thus damaging his/her privacy. Furthermore, it is important to be cautious of the results of any deblurring system as failures could cause misjudgment. For example, the inaccurate restoration of numbers and letters can produce misleading information. Our proposed approach is robust to various noise levels and inaccurate kernels, which intuitively improves its adaptability to more complex scenes and thus minimizes the chance of such failures. Nevertheless, misjudgment based on incorrect restoration cannot be ruled out completely.

## Acknowledgments and Disclosure of Funding

This project has received funding from the European Research Council (ERC) under the European Union's Horizon 2020 research and innovation programme (grant agreement No. 866008). We thank Uwe Schmidt for helpful feedback on the manuscript.

## Footnotes

[1]Note that deconvolution is a particular way of solving image deblurring given the blur model in Eq. (1).

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
