[Supplementary Material]

# Deep Wiener Deconvolution: Wiener Meets Deep Learning for Image Deblurring
## – Supplemental Material –

**Jiangxin Dong**
MPI Informatics
jdong@mpi-inf.mpg.de

**Stefan Roth**
TU Darmstadt
stefan.roth@visinf.tu-darmstadt.de

**Bernt Schiele**
MPI Informatics
schiele@mpi-inf.mpg.de

## Overview

In this supplemental material, we first present the detailed network architecture and parameters of the proposed approach in Sec. A. We further provide more analysis of the proposed method and ablation studies in Sec. B. Section C shows some qualitative results for potential applications of the proposed approach on medical imaging and imaging in astronomy. We provide more visual comparisons against other methods in Sec. D.

## A  Configurations of the Deep Wiener Deconvolution Network

As shown in Fig. 2 of the main paper, our proposed network consists of a feature extraction network, a feature-based Wiener deconvolution, and a multi-scale feature refinement network. Tables 6 and 7 list the detailed configurations of the feature extraction network and the feature refinement network for one scale, respectively. Following Eq. (9) of the main paper, the number of input channels in the feature refinement network for scale $l = 1$ and $l > 1$ differs. Hence, our feature refinement network shares its parameters across all scales except for the first encoder block.

Table 6: Parameters of the feature extraction network. *Conv* denotes a convolutional layer and *Res* denotes a residual block.

| Layers | $\text{Conv}_1$ | $\text{Res}_1$ | $\text{Res}_2$ | $\text{Res}_3$ |
|---|---|---|---|---|
| Filter size | 5 | 5 | 5 | 5 |
| Number of filters | 16 | 16 | 16 | 16 |
| Stride | 1 | 1 | 1 | 1 |

## B  In-Depth Analysis

In this section, we provide additional detailed ablation studies and discussions on the proposed deep Wiener deconvolution network.

### B.1  Effect of the feature-based Wiener deconvolution with a basic reconstruction network

Our goal of this section is to complement the results in the main paper (in particular Tab. 3 in the main paper) and to analyze the effect of the feature information on the classical Wiener deconvolution.

Table 7: Parameters of the feature refinement network. *Conv* denotes a convolutional layer, *Res* denotes a residual block, and *Deconv* denotes a transposed deconvolutional layer. $Res_i$–$Res_{i+2}$ (i=1, 4, 7, 10, 13, 16) represents a series of three identically configured residual blocks.

| Layers | $Conv_1$ | $Res_1$–$Res_3$ | $Conv_2$ | $Res_4$–$Res_6$ | $Conv_3$ | $Res_7$–$Res_9$ |
|---|---|---|---|---|---|---|
| Filter size | 5 | 5 | 5 | 5 | 5 | 5 |
| Number of filters | 32 | 32 | 64 | 64 | 128 | 128 |
| Stride | 1 | 1 | 2 | 1 | 2 | 1 |

| Layers | $Res_{10}$–$Res_{12}$ | $Deconv_1$ | $Res_{13}$–$Res_{15}$ | $Deconv_2$ | $Res_{16}$–$Res_{18}$ | $Conv_4$ |
|---|---|---|---|---|---|---|
| Filter size | 5 | 3 | 5 | 3 | 5 | 5 |
| Number of filters | 128 | 64 | 64 | 32 | 32 | 3 |
| Stride | 1 | 2 | 1 | 2 | 1 | 1 |

Table 8: Effect of the feature-based Wiener deconvolution. All methods are evaluated on the datasets of [21] (1% noise level) and [24] (1% noise level). The basic reconstruction network that contains three residual blocks followed by one convolutional layer is denoted as *Basic reconstruction*. The baseline method that follows the classical Wiener deconvolution with the Basic reconstruction network is denoted as $Wiener_I$. The baseline methods that combine the classical Wiener deconvolution with the image gradients, the concatenation of both the input and the image gradients, 9 learned deep features, and 16 learned deep features are denoted as $Wiener_G$, $Wiener_{I+G}$, $Wiener_{9\,features}$, and $Wiener_D$, respectively.

| | Feature extraction | | | Wiener deconvolution | Refinement / Reconstruction | | PSNR in dB/SSIM | |
|---|---|---|---|---|---|---|---|---|
| | Intensity | Gradients | Deep features | | Basic reconstruction | Multi-scale refinement | [21] | [24] |
| Wiener [49] | ✔ | ✗ | ✗ | ✔ | ✗ | ✗ | 27.48/0.6981 | 26.76/0.7033 |
| $Wiener_I$ | ✔ | ✗ | ✗ | ✔ | ✔ | ✗ | 30.32/0.8931 | 27.14/0.8226 |
| $Wiener_G$ | ✗ | ✔ | ✗ | ✔ | ✔ | ✗ | 18.03/0.6456 | 13.46/0.5350 |
| $Wiener_{I+G}$ | ✔ | ✔ | ✗ | ✔ | ✔ | ✗ | 33.37/0.9239 | 29.35/0.8423 |
| $Wiener_{9\,features}$ | ✗ | ✗ | ✔ | ✔ | ✔ | ✗ | 34.09/0.9394 | 30.46/0.8755 |
| $Wiener_D$ | ✗ | ✗ | ✔ | ✔ | ✔ | ✗ | 34.78/0.9478 | 30.95/0.8836 |
| Ours | ✗ | ✗ | ✔ | ✔ | ✗ | ✔ | **36.90/0.9614** | **31.74/0.8938** |

We thus disable our proposed feature refinement network for all the baseline methods in this section. We compare the classical Wiener deconvolution [49] with various baseline methods that combine the classical Wiener deconvolution with various features: only the image gradients along the vertical and horizontal directions ($Wiener_G$ for short), the concatenation of both the blurry image and the image gradients ($Wiener_{I+G}$ for short), and learned deep features ($Wiener_{9\,features}$ and $Wiener_D$ for short, latter using 16 features). To reconstruct the final clear image from the deconvolved features, we use a basic reconstruction network that contains three residual blocks followed by one convolutional layer. Note that this is the same basic reconstruction network discussed in the main paper. For fair comparison, we further compare with a baseline method that follows the classical Wiener deconvolution in image space with the same reconstruction network ($Wiener_I$ for short). Table 8 shows that using the gradient information alone is not sufficient, but is effective to improve the deconvolution performance when combined with the intensity information from the blurry image. Thus, combining useful feature information is able to improve the performance of the classical Wiener deconvolution. Furthermore, the baseline method with the learned deep features performs the best among all the baseline methods, increasing the PSNR by at least 0.72dB on the dataset by [21] and 1.11dB on the dataset by [24]. The results in Tab. 8 demonstrate that the deep features are more effective for extracting useful information for better deblurring and do not require a manual feature combination compared to fixed feature extractors.

The last row in Tab. 8 reports the results of the proposed approach with the deep Wiener deconvolution module and our multi-scale feature refinement module, which demonstrates that the multi-scale refinement network is much better at recovering the clear image from the deconvolved features, increasing the PSNR by 2.12dB on the dataset by [21] and 0.79dB on the dataset by [24].

Table 9: Effectiveness of the deep features learned by our piece-wise linear feature extraction network, evaluated on the datasets of [21], [24], and [43] (PSNR in dB/SSIM). We compare the baseline method that combines the classical Wiener deconvolution with the proposed deep feature extraction network ($Wiener_D$) with a baseline that uses a linear feature extraction network ($Wiener_D$ w/ learned linear features).

| Dataset | Noise level | $Wiener_D$ w/ learned linear features | $Wiener_D$ |
|---|---|---|---|
| Levin et al. [21] | 1% | 33.69/0.9133 | **34.78/0.9478** |
| | 3% | 29.80/0.8343 | **31.18/0.8942** |
| | 5% | 28.09/0.7888 | **29.39/0.8553** |
| BSDS [24] | 1% | 31.69/0.8815 | **32.69/0.9127** |
| Sun et al. [43] | 1% | 29.46/0.8472 | **30.95/0.8836** |

Table 10: Quantitative comparison to state-of-the-art methods on the dataset of Levin et al. [21].

| Noise level | | DMPHN [53] | EPLL [56] | MLP [40] | CSF [37] | LDT [8] | IRCNN [55] | FNBD [42] | RGDN [10] | Ours |
|---|---|---|---|---|---|---|---|---|---|---|
| 15% | PSNR (dB) | 21.35 | 22.04 | 22.98 | 21.87 | 21.46 | 24.64 | 21.75 | 17.10 | **25.57** |
| | SSIM | 0.5144 | 0.5987 | 0.5562 | 0.5130 | 0.4309 | 0.6647 | 0.4411 | 0.2491 | **0.7429** |
| 20% | PSNR (dB) | 20.45 | 21.13 | 22.13 | 20.82 | 21.96 | 22.68 | 20.21 | 14.81 | **24.01** |
| | SSIM | 0.4301 | 0.5579 | 0.5350 | 0.4618 | 0.5177 | 0.5912 | 0.3508 | 0.1599 | **0.6765** |
| 30% | PSNR (dB) | 18.66 | 19.28 | 20.32 | 13.15 | 19.15 | 20.97 | 18.06 | 12.34 | **21.79** |
| | SSIM | 0.2909 | 0.4457 | 0.4916 | 0.1103 | 0.3124 | 0.5554 | 0.2378 | 0.0889 | **0.5637** |

Table 11: Quantitative comparison to state-of-the-art methods on the dataset of Martin et al. [24].

| Noise level | | DMPHN [53] | EPLL [56] | MLP [40] | CSF [37] | LDT [8] | FCN [54] | IRCNN [55] | FDN [16] | FNBD [42] | RGDN [10] | Ours |
|---|---|---|---|---|---|---|---|---|---|---|---|---|
| 3% | PSNR (dB) | 24.04 | 26.28 | 25.62 | 26.33 | 26.24 | 26.92 | 27.18 | 27.23 | 27.44 | 27.06 | **28.58** |
| | SSIM | 0.6538 | 0.6996 | 0.6505 | 0.7096 | 0.7018 | 0.7346 | 0.7219 | 0.7505 | 0.7618 | 0.7620 | **0.8040** |
| 5% | PSNR (dB) | 23.72 | 24.66 | 24.01 | 24.93 | 24.90 | 25.45 | 25.65 | 25.93 | 25.49 | 25.33 | **27.29** |
| | SSIM | 0.6255 | 0.6276 | 0.5619 | 0.6428 | 0.6358 | 0.6771 | 0.6640 | 0.6943 | 0.6589 | 0.6688 | **0.7573** |

## B.2 Effectiveness of learned deep features with a basic reconstruction network

In this section, we focus on the effect of learned deep features and disable our proposed feature refinement network for all the baseline methods. To reconstruct the final image from the deconvolved deep features, we use the same basic reconstruction network as in Tab. 8. As stated in the main paper, the derivations of the feature-based Wiener deconvolution strictly hold in a linear feature space. Moreover, we are interested to leverage powerful learned feature extractors $\{\mathbf{F}_i\}$ beyond hand-crafted ones. To this end, we develop a feature extraction network to estimate $\{\mathbf{F}_i \mathbf{y}\}$. As the feature extraction network with ReLUs is piece-wise linear, the linearity assumption of the Wiener deconvolution holds only locally [19, 27]. To evaluate the feasibility and effectiveness of this piece-wise linear feature extraction network, we remove the ReLUs in the proposed feature extraction network and compare the effect of using features extracted by the proposed network ($Wiener_D$ for short) and the linear feature extraction network ($Wiener_D$ w/ learned linear features for short) on the classical Wiener deconvolution in Tab. 9. The results show that the method with the features extracted by the proposed piece-wise linear (deep) feature extraction network performs much better than that using only a linear feature extraction network, increasing the PSNR by at least 1.09dB on the dataset of [21], 1.00dB on the dataset of [24], and 1.49dB on the dataset of [43].

## B.3 More experimental results on blurry images with higher noise levels

In the main paper, we compare the proposed method with state-of-the-art methods on the dataset of Levin et al. [21] with Gaussian noise of 1%–10%, and on the datasets of Martin et al. [24] and Sun et al. [43] with Gaussian noise of 1%. In this section, we further evaluate our method on these test datasets with higher levels of Gaussian noise. Tables 10 and 11 show that the proposed method performs better than state-of-the-art methods on blurry images also with higher noise levels.

Figure 6: Illustration of learned deep features. *(a)* Blurry input. *(b)* Ground truth. *(c)–(e)* are the visualization of one channel of the blurry input, the image gradient along the vertical direction, and the image gradient along the horizontal direction, respectively. *(f)–(h)* are the deconvolved results corresponding to *(c)–(e)*. *(i)* and *(j)* visualize some features learned by the feature extraction network and the corresponding deconvolved results, respectively.

## B.4 Visualization of learned deep features

To intuitively illustrate what the proposed feature extraction network learns, we show some learned features in Fig. 6(i), where the corresponding results deconvolved by Eqs. (3) and (8) of the main paper are shown in Fig. 6(j). The blurry input and ground truth are shown in Fig. 6(a)–(b). For better understanding, we also show the visualization of one channel of the blurry input and the image gradients along two directions in Fig. 6(c)–(e). Their corresponding deconvolved results are shown in Fig. 6(f)–(h). Compared to the intensity information and the gradient information in Fig. 6(c)–(h), the learned deep features in Fig. 6(i)–(j) contain much richer feature information, facilitating the reconstruction of high-quality images. In addition, by integrating the feature extraction, the deep Wiener deconvolution, and the feature refinement into an end-to-end network, the feature extraction network can automatically learn useful feature information from the blurry input for better deblurring. Thus, the proposed method does not require a manual combination of features.

## B.5 Role of feature extraction network

As stated in the main paper, we propose to obtain powerful feature extractors $\{\mathbf{F}_i\}$ using deep neural networks that provide more useful information for a subsequent Wiener deconvolution. However, on may actually wonder whether the feature extraction network acts as a denoiser, leading to the observed robustness of the proposed method to various noise levels. To answer this question, we further compare the learned features for blurry images without and with noise of different noise levels $(1\%, 5\%, 10\%)$ in Fig. 7. The results show that the higher the noise level of the blurry image is, the more noise the learned feature contains. Therefore, the feature extraction network does not appear to act as a denoiser.

## B.6 Robustness to differing noise levels

**Estimation of parameters $\{s_i^x\}$ and $\{s_i^n\}$.** As mentioned in the main paper, $s_i^x$ and $s_i^n$ denote $\mathbb{E}(|\mathbf{F}_i\mathbf{x}|^2)$ and $\mathbb{E}(|\mathbf{F}_i\mathbf{n}|^2)$, respectively. However, in real applications, it is hard to accurately calculate

Blurry input          Features learned by the proposed feature extraction network

(a) Results on the blurry input without noise

(b) Results on the blurry input with $1\%$ Gaussian noise

(c) Results on the blurry input with $5\%$ Gaussian noise

(d) Results on the blurry input with $10\%$ Gaussian noise

Figure 7: Learned features for blurry images with noise of different levels. From *(a)* to *(d)*, the noise level in the blurry input becomes higher and the extracted features contain more noise.

these expectations. Similar to existing methods [42, 51], we estimate $s_i^x$ by the standard deviation of the blurry feature $\mathbf{F}_i \mathbf{y}$. $s_i^n$ is estimated by the variance of the difference between the blurry feature $\mathbf{F}_i \mathbf{y}$ and the mean-filtered result of $\mathbf{F}_i \mathbf{y}$. Note that $s_i^n$ is adaptively computed from the blurry feature $\mathbf{F}_i \mathbf{y}$ and related to the noise level in $\mathbf{F}_i \mathbf{y}$. Thus, the proposed network is able to handle blurry images with various noise levels. This is also referred to as being noise-blind [13]. To demonstrate the robustness of the proposed deep Wiener deconvolution network to various noise levels, we compare our *single* model (which is trained with mixed Gaussian noise levels, ranging from 0–5%) with instances of our model that are specifically trained with only one Gaussian noise level of either $1\%$, $3\%$, or $5\%$. Table 12 shows that the proposed noise-blind model obtains similar results to the noise-specific models across various noise levels, e.g., 36.90dB compared to 37.02dB for $1\%$ Gaussian noise and 30.77dB compared to 30.80dB for $5\%$ Gaussian noise.

As mentioned in Section 4.1 of the main paper, our normal training dataset contains blurry images with Gaussian noise of various noise levels, ranging from 0 to $5\%$. One may thus wonder whether the robustness of the proposed model is due to various noise levels in the training dataset and whether the

Table 12: Robustness of the proposed deep Wiener devoncolution network to various noise levels, evaluated on the dataset of Levin et al. [21] (PSNR in dB/SSIM).

| Noise level | Baseline model trained on images with specific noise level | Ours (single model) |
|---|---|---|
| 1% | **37.02/0.9624** | 36.90/0.9614 |
| 3% | **32.88/0.9191** | 32.77/0.9179 |
| 5% | **30.80**/0.8846 | 30.77/**0.8857** |
| 10% | **28.04/0.8241** | 27.78/0.8189 |

proposed model is still effective when the noise level is outside of this range. To answer this question, we evaluate the proposed model on Levin's dataset [21] with $10\%$ Gaussian noise, a noise level that is not included in our training dataset. Table 12 shows that the proposed model still performs close to a noise-specific model trained on images with $10\%$ Gaussian noise. In practice, this noise robustness thus allows us to avoid training noise-specific models.

We further carry out a sensitivity analysis w.r.t. the estimation of the parameters $\{\frac{s_i^n}{s_i^x}\}$. We use the dataset of [21] with $5\%$ Gaussian noise and add $0$–$20\%$ perturbation to our estimated $\{\frac{s_i^n}{s_i^x}\}$ (no retraining). Figure 8 shows that the PSNR values differ no more than $0.06$dB, suggesting the robustness of our method to the estimation of the parameters $\{\frac{s_i^n}{s_i^x}\}$.

Figure 8: Sensitivity analysis w.r.t. the estimation of $\{\frac{s_i^n}{s_i^x}\}$.

The robustness of the proposed method to various noise levels not only stems from the estimation of the parameters $\{\frac{s_i^n}{s_i^x}\}$, but also from our end-to-end network. Such benefit has also been demonstrated in high-level computer vision tasks. Diamond et al. [6] propose an end-to-end architecture for joint denoising, deblurring, and classification, which makes classification more robust to realistic noise and blur. By embedding the Wiener deconvolution into an end-to-end network, our proposed method facilitates a feature extractor for learning useful features for deconvolution with fewer artifacts. Our architecture also benefits from feature refinement of the deconvolved features to reconstruct clearer images.

## B.7   Choice of parameters $\{\frac{s_i^n}{s_i^x}\}_{i=1}^{M}$

In the proposed feature-based Wiener deconvolution of Eqs. (3) and (8) of the main paper, the value of $\frac{s_i^n}{s_i^x}$ is used to control the quality of the deconvolved feature $\mathbf{F}_i\hat{\mathbf{x}}$ and estimated from each blurry feature $\mathbf{F}_i\mathbf{y}$ as described in Sec. B.6. An alternative would be to estimate a single value from the

blurry input $\mathbf{y}$ for all the parameters $\{\frac{s_i^n}{s_i^x}\}_{i=1}^M$. However, the properties of the extracted feature $\mathbf{F}_i\mathbf{y}$ may be quite different from that of the blurry input $\mathbf{y}$. For example, the noise level of the extracted feature $\mathbf{F}_i\mathbf{y}$ can be lower than that of the blurry input $\mathbf{y}$. Then using the parameter estimated from $\mathbf{y}$, fine-scale structures and detail can be over-smoothed in the deconvolved feature $\mathbf{F}_i\mathbf{x}$. Table 13 shows that estimating each $\frac{s_i^n}{s_i^x}$ from the corresponding blurry feature $\mathbf{F}_i\mathbf{y}$ is more effective than a baseline model with a single parameter estimated from the blurry input $\mathbf{y}$, increasing the PSNR about $0.66$dB on the dataset of [21] with $1\%$ Gaussian noise.

Table 13: Ablation study of the parameters $\{\frac{s_i^n}{s_i^x}\}_{i=1}^M$ (PSNR in dB/SSIM).

| Dataset | Noise level | Baseline model with a single value for all $\{\frac{s_i^n}{s_i^x}\}_{i=1}^M$ | Proposed model |
|---|---|---|---|
| Levin et al. [21] | 1% | 36.24/0.9553 | **36.90/0.9614** |
| | 3% | 32.44/0.9126 | **32.77/0.9179** |
| BSDS [24] | 1% | 31.51/0.8890 | **31.74/0.8938** |

## B.8 Number of extracted deep features

The proposed approach learns 16 features to extract useful information from the blurry input for the Wiener deconvolution module. We further evaluate the effect of the number of deep features by varying the number of learned features from 9 to 64. Table 14 shows that using more features improves the image quality, however with diminishing returns. We empirically use 16 features as a trade-off between image quality and efficiency.

Table 14: Ablation study on the number of deep features on the dataset of Levin et al. [21] (PSNR in dB/SSIM).

| Noise level / Number of features | 9 | 16 | 32 | 64 |
|---|---|---|---|---|
| 1% | 36.81/0.9608 | 36.90/0.9614 | 36.95/0.9615 | **36.96/0.9619** |
| 3% | 32.73/0.9177 | 32.77/0.9179 | 32.83/0.9180 | **32.85/0.9191** |
| 5% | 30.72/0.8854 | 30.77/0.8857 | 30.80/0.8854 | **30.89/0.8884** |

## B.9 Batch size

We use a batch size of 8 for training. We further evaluate the effect of the batch size by varying it from 1 to 16 in Tab. 15.

Table 15: Ablation study on the batch size (PSNR in dB/SSIM).

| Dataset / Batch size | 1 | 4 | 8 | 16 |
|---|---|---|---|---|
| Levin et al. [21] | 36.73/0.9601 | **36.90**/0.9610 | **36.90/0.9614** | 36.81/0.9609 |
| BSDS [24] | 31.49/0.8929 | 31.65/0.8932 | **31.74/0.8938** | 31.69/0.8926 |

## B.10 Number of feature refinement scales

In the main paper, we evaluate the effectiveness of the multi-scale strategy in the feature refinement module and use a total of two scales. We further analyze whether using more scales will improve the image quality. Table 16 demonstrates that using more scales does not significantly improve the image quality. We empirically use 2 scales as a trade-off between image quality and efficiency.

## B.11 Loss weights for different scales

To better regularize the proposed network, we apply an $\ell_1$ loss function at each scale $l$, as defined in Eq. (10) of the main paper. We show the effect of the weights $\{\gamma_l\}$ for different scales in Tab. 17. We

Table 16: Ablation study on the number of feature refinement scales (PSNR in dB/SSIM).

| Noise level | Method / Kernel size | $13 - 19$ | $33 - 39$ | $53 - 59$ |
|---|---|---|---|---|
| 1% | w/o multi-scale | 33.46/0.9232 | 29.23/0.8398 | 27.45/0.7995 |
| | Ours w/ 2 scales | 33.57/0.9254 | 29.47/0.8437 | 27.73/0.8127 |
| | Ours w/ 3 scales | 33.61/0.9263 | 29.48/0.8460 | 27.96/0.8160 |
| 3% | w/o multi-scale | 29.93/0.8475 | 26.71/0.7370 | 25.20/0.6918 |
| | Ours w/ 2 scales | 30.09/0.8519 | 27.00/0.7510 | 25.57/0.7080 |
| | Ours w/ 3 scales | 30.09/0.8526 | 27.03/0.7523 | 25.69/0.7092 |

note that the proposed method is less effective when $\gamma_2 < \gamma_1$. Since the output of the scale 2 is the desired one, it is more reasonable to set $\gamma_2$ to a larger value. Table 17 demonstrates that the results are robust when $\gamma_2 \geq \gamma_1$. We empirically set $\gamma_2 = \gamma_1 = 1$.

Table 17: Ablation study on the loss weights for different scales (PSNR in dB/SSIM).

| $(\gamma_1, \gamma_2)$ | $(0.2, 1)$ | $(0.5, 1)$ | $(1, 1)$ | $(1, 0.5)$ |
|---|---|---|---|---|
| Levin et al. [21] | **36.97**/0.9612 | 36.86/0.9608 | 36.90/**0.9614** | 36.78/0.9596 |
| BSDS [24] | 31.71/**0.8940** | 31.72/0.8937 | **31.74**/0.8938 | 31.53/0.8890 |

## B.12 Additional reconstruction module

Our method consists of a deep Wiener deconvolution module to generate deconvolved features and then relies on a multi-scale feature refinement module to refine the latent features and reconstruct the final clear image. To understand the potential benefit of using an additional reconstruction module, we compare with a variant that adopts an additional reconstruction module at the end of the network that consists of three residual blocks followed by one convolutional layer to obtain the final clear image. Table 18 shows that using an additional reconstruction module does not significantly improve the accuracy. Considering the efficiency, our proposed architecture does not adopt an additional reconstruction module.

Table 18: Ablation study on an additional reconstruction module (PSNR in dB/SSIM).

| Dataset | Baseline model with an additional reconstruction module | Ours |
|---|---|---|
| Levin et al. [21] | **36.98/0.9615** | 36.90/0.9614 |
| BSDS [24] | 31.73/0.8936 | **31.74/0.8938** |

## B.13 Extension to non-uniform image deblurring

Our method is derived based on a uniform blur model. However, our method can be extended to handle non-uniform blur by applying the local uniform approximation method of [48]. We evaluate the effectiveness of our method on non-uniform image deblurring in Fig. 9, where the blurry examples are from [59] and the blur kernels are estimated by the method of [60]. The result by [60] over-smoothes the detail in the restored images, as shown in Fig. 9 (b). In contrast, our method generates much clearer images in Fig. 9 (c), where the fine-scale structures of the books and branches are recovered much better.

## B.14 Robustness to outliers

As demonstrated in Sec. B.6, the proposed method is robust to different noise levels. We further explore the applicability of our method to images with outliers [1, 58]. As shown in Fig. 10 on blurry images with impulse noise and saturated areas, without fine-tuning on images with outliers, the proposed method performs comparably with a method that is specifically designed to handle outliers [58].

However, for blurry input with significant outliers in Fig. 11, directly using the proposed model trained on images with Gaussian noise will generate a deblurred result with severe artifacts. However,

(a) Blurry input        (b) [60]        (c) Ours

Figure 9: Examples with non-uniform image deblurring. The images shown in *(b)* are obtained from the reported results of [60]. Compared to the results in *(b)*, the images recovered by our method in *(c)* are much clearer with finer detail.

(a) Blurry input     (b) FNBD [42]     (c) [58]     (d) Ours

Figure 10: Robustness to outliers. *(a)* Above: blurry input with impulse noise. Below: blurry input with impulse noise and saturated areas. The proposed method without fine-tuning can achieve comparable results against a method specifically designed for outlier handling [58].

we can refine the proposed network on the same dataset described in Section 4.1 of the main paper by adding impulse noise ranging from 0–5%. The such refined model can generate a much clearer image with finer detail, see Fig. 11(d), compared to the result obtained by [58] in Fig. 11(b).

## B.15    Run-time

We benchmark the run-time of a selection of evaluated methods on a machine with an Intel Xeon E5-2650 v4 CPU and an NVIDIA TITAN Xp GPU. Table 19 summarizes the average run-time of representative methods. Our deep Wiener deconvolution network requires about $0.05$ seconds on the images (with $255 \times 255 \times 3$ pixels) from Levin et al. [21] and roughly $0.40$ seconds on images (with $800 \times 1024 \times 3$ pixels) from Sun et al. [43]. The methods [10, 16, 55] are iterative and thus take more time. The approach of [42] contains a postprocessing step of solving an optimization problem, which also takes a certain amount of time. In contrast, the proposed deep Wiener deconvolution network

| (a) Blurry input | (b) [58] | (c) Ours | (d) Ours w/ finetuning |

Figure 11: An example with significant impulse noise.

is based on an end-to-end architecture, which does not require iterative solutions or postprocessing steps. Thus, the proposed approach runs faster with high image quality.

Table 19: Run-time performance (seconds). All the methods are evaluated on the same machine with the same settings.

| Image size | IRCNN [55] | FDN [16] | FNBD [42] | RGDN [10] | Ours |
|---|---|---|---|---|---|
| $255 \times 255 \times 3$ | 0.32 | 0.60 | 0.07 | 1.44 | 0.05 |
| $800 \times 1024 \times 3$ | 4.59 | 1.85 | 0.67 | 11.27 | 0.40 |

# C  Potential Applications

As discussed in the Broader Impact section of the main paper, image deblurring has a broad potential impact through a wide range of applications, e.g., medical imaging, telescope imaging in astronomy, portable device imaging, etc. Our image deblurring method based on the proposed deep Wiener deconvolution network can provide high-quality clear images to facilitate intelligent data analysis tasks in these fields. To illustrate its applicability, in this section, we provide some examples for potential applications of our approach to medical imaging and astronomy imaging. The results are obtained without finetuning of our model to these particular images or domains.

(a) Blurry image        (b) Ours

Figure 12: Example with a real point spread function from [57]. The blur kernel is estimated by the method of [29]. The result is obtained with the proposed model without fine-tuning, which demonstrates the potential application of our approach to medical imaging.

(a) Blurry image        (b) Ours

Figure 13: Example with a real point spread function from the Internet. The blur kernel is estimated by the method of [29]. The result is obtained with the proposed model without fine-tuning, which demonstrates the potential application of our approach to telescope imaging in astronomy.

# D    Qualitative Comparisons

In this section, we present additional visual comparisons with the state-of-the-art methods [8, 10, 16, 37, 40, 42, 53–56] on images with both simulated (Figs. 14 to 19) and real-world blur (Figs. 20 to 22).

Figure 14: Example with simulated blur (1% noise level) from the dataset of [21]. The deblurred results *(b)–(j)* by existing methods have obvious visual distortion in the shirt as enclosed in the red boxes and the detail is over-smoothed in the sweater as enclosed in the yellow boxes. In contrast, our deep Wiener deconvolution network can generate a much clearer image with finer detail and fewer artifacts.

(a) Blurry image    (b) EPLL [56]    (c) MLP [40]    (d) CSF [37]

(e) LDT [8]    (f) FCN [54]    (g) IRCNN [55]    (h) FDN [16]

(i) FNBD [42]    (j) RGDN [10]    (k) Ours    (l) Ground truth

Figure 15: Example with simulated blur (3% noise level) from the dataset of [21]. The method [40] is less effective in generating a clear result as shown in *(c)*. Existing methods can remove the blur and noise, but some detail is also smoothed as shown in the red and yellow boxes of *(b)* and *(d)–(j)*. However, our approach restores a much clearer image with finer detail and fewer artifacts.

Figure 16: Example with simulated blur (5% noise level) from the dataset of [21]. The result obtained by [40] has severe artifacts in *(c)*. For other methods, small-scale structures and detail are over-smoothed as shown in the red and yellow boxes of *(b)* and *(d)–(j)*. Compared to existing methods, our approach can effectively preserve finer detail as shown in *(k)*.

Figure 17: Example with simulated blur (10% noise level) from the dataset of [21]. The results by competing methods have severe artifacts in *(b)–(j)*. In contrast, the proposed deep Wiener deconvolution network is able to generate a clear result *(k)* from the blurry image despite significant image noise.

(a) Blurry image  (b) EPLL [56]  (c) MLP [40]  (d) CSF [37]

(e) LDT [8]  (f) FCN [54]  (g) IRCNN [55]  (h) FDN [16]

(i) FNBD [42]  (j) RGDN [10]  (k) Ours  (l) Ground truth

Figure 18: Example with simulated blur (1% noise level) from the dataset of [24]. Compared to the results in *(b)–(j)*, the proposed method is able to effectively preserve detail and fine-scale structures as shown in *(k)*.

(a) Blurry image    (b) EPLL [56]    (c) MLP [40]    (d) CSF [37]

(e) LDT [8]    (f) FCN [54]    (g) IRCNN [55]    (h) FDN [16]

(i) FNBD [42]    (j) RGDN [10]    (k) Ours    (l) Ground truth

Figure 19: Example with simulated blur (1% noise level) from the dataset of [43]. The result by [42] in *(i)* has severe artifacts. For other results, small-scale structures and detail are over-smoothed. In contrast, our method is more effective in restoring the characters.

(a) Blurry image      (b) EPLL [56]      (c) CSF [37]

(d) LDT [8]      (e) FCN [54]      (f) IRCNN [55]

(g) FNBD [42]      (h) RGDN [10]      (i) Ours

Figure 20: Example with real camera shake from [30]. The blur kernel in *(a)* is estimated by the method of [30]. The result obtained by [42] has severe artifacts *(g)*. Compared to existing methods, our deep Wiener deconvolution network is able to preserve finer-scale detail *(i)*.

(a) Blurry image             (b) EPLL [56]

(c) MLP [40]             (d) CSF [37]

(e) LDT [8]             (f) FCN [54]

(g) IRCNN [55]             (h) FNBD [42]

(i) RGDN [10]             (j) Ours

Figure 21: Example with real camera shake from [50]. The blur kernel in *(a)* is estimated by the method of [29]. The deblurred results by [40, 42] have severe artifacts as shown in the yellow and red boxes of *(c)–(h)*. For other methods, fine-scale structures and detail are over-smoothed. Compared to existing methods, our method recovers much clearer characters as shown in *(j)*.

(a) Blurry image     (b) EPLL [56]

(c) CSF [37]     (d) FCN [54]

(e) IRCNN [55]     (f) FDN [16]

(g) RGDN [10]     (h) Ours

Figure 22: Example with real camera shake from [50]. The blur kernel in *(a)* is estimated by the method of [29]. The result in *(a)* has severe artifacts, while for the other results, fine-scale structures are over-smoothed. In contrast, our approach generates an image with much clearer characters.