[Reviews · NeurIPS 2020]

Review 1

Summary and Contributions: This paper tackles the problem non-blind image deconvolution using a deep-learning approach inspired by the popular Wiener filter. The main idea is to incorporate the blur kernel information in the form of a Wiener filter that is applied to extracted features instead of directly to the input image pixels. Then, the deconvolved features are processed by a multi-scale network that predicts the final image. The method produces high-quality results both in terms of visual quality and quantitative results. Image deblurring, and in particular non-blind image deconvolution is a longstanding problem in image processing and computer vision. This paper introduces an interesting approach by combining classical ideas (Wiener filter) in a deep learning modern approach.

Strengths: Introduce a method for non-blind image deconvolution that leverages the information regarding the blur kernel in a sound way. Extensive experiments show that the method outperforms the competitors. The evaluation is done in a different dataset with images having different noise levels. The paper presents an ablation study that shows how all the introduced components are required. Also it is shown that learning the feature space where the Wiener deconvolution takes place produces an improvement over using hand-crafted features (e.g., gradient features).

Weaknesses: The paper doesn't introduce a formal justification why it makes sense to deblur the extracted features (that are non-linear, so the Wiener deduction can't be applied). This is just presented as a way to motivate the approach. Experimental results show low to moderate noise levels (1%-10% in one dataset and ). Will the noise level estimation work at higher noise levels? Also, what happens if the blur is non-uniform, or the linear model assumption fails (e.g., gamma correction, non-linear CRF). Will the proposed method introduce artifacts? This is not discussed in the paper.

Correctness: The claims seem correct. The empirical evaluation seems reasonable to validate the claims.

Clarity: The paper is well written.

Relation to Prior Work: It would be interesting to discuss the connection to other approaches that rely on kernel estimation information to augment the processing somewhere in the middle of the processing network:, for example: Diamond, S., Sitzmann, V., Boyd, S., Wetzstein, G. and Heide, F., 2017. Dirty pixels: Optimizing image classification architectures for raw sensor data. arXiv preprint arXiv:1701.06487. The paper makes an extensive analysis of the previous work. To complement the analysis I recommend including papers that tackle the problem of non-blind deconvolution in a more general setting where compression artifacts, or other non-linearities may be present. The work by Xu et al. [45] is already cited, but this is not commented as one of the advantages of the approach. On the same line, but following a more classical variational method (see the evaluation setup to get ideas for testing the proposed method): Anger, J., Facciolo, G. and Delbracio, M., 2018, October. Modeling Realistic Degradations in Non-Blind Deconvolution. In 2018 25th IEEE International Conference on Image Processing (ICIP) (pp. 978-982). IEEE. Additionally, It would be interesting to see more comparisons regarding other Plug-n-play methods, such as: Romano, Y., Elad, M. and Milanfar, P., 2017. The little engine that could: Regularization by denoising (RED). SIAM Journal on Imaging Sciences, 10(4), pp.1804-1844. E. Ryu, J. Liu, S. Wang, X. Chen, Z. Wang, and W. Yin, Plug-and-play methods provably converge with properly trained denoisers, International Conference on Machine Learning (ICML), Long Beach, CA, 2019.

Reproducibility: Yes

Additional Feedback: AFTER REBUTTAL The authors did a good job answering many of the raised points. I think this is a good submission and therefore I'm keeping my acceptance score (7). ----- Other comments: -- Deblurring is not a synonym of deconvolution. I believe these two terms are used interchangeably all over the paper. Please consider mentioning somewhere that deconvolution is a particular way of doing deblurring. --L138 - Actually, I wouldn't say "it is reasonable to apply" since without the linearity the whole deduction is not valid (even if the network can be seen as piecewise linear). -- Have you considered sharing the network parameters for all the scales? -- Why PSNR values in Table 5 are so different from the ones in Table 3. Could you add the information regarding each experiment (in particular the noise level, dataset, kernel dataset). -- The gain regarding learning the features is marginal (+0.4db) with respect to other components in the pipeline. Could you elaborate a little more on this.


Review 2

Summary and Contributions: This paper proposes a feature-based Wiener deconvolution module to explore useful learned features for image deblurring. Besides a multi-scale feature refinement module is proposed to restore fine-scale detail from the deconvolved features.

Strengths: 2. 1. This paper uses Wiener deconvolution in the feature space and show the superior compared to directly performing on the image space. 2. This paper proposes a multi-scale feature refinement module to restore the fine-scale structures of the deconvolved features, facilitating the reconstruction of high-quality images. 3. comparisons with state-of-the-art methods demonstrate that the proposed approach perform well and is robust to different noise levels.

Weaknesses: 1.The Wiener deconvolution has been proposed already, thus the main contribution is not much sufficient. 2. The motivation of using the Wiener deconvolution is not introduced completely.

Correctness: yes

Clarity: yes

Relation to Prior Work: yes

Reproducibility: Yes

Additional Feedback: see the weakness


Review 3

Summary and Contributions: This paper presents a deep neural network model for novel non-blind deconvolution that is based on the Wiener deconvolution. The network consists of two parts: the feature-based wiener deconvolution step, and the multi-scale feature refinement step. The feature-based wiener deconvolution step extends the traditional wiener deconvolution to the feature space, i.e., it embeds the input image into the feature space and performs the deconvolution in the feature space. The deconvolved features are then refined in the multi-scale refinement step. The entire process is implemented as a single end-to-end network, and all the network parameters are learned in an end-to-end manner. The paper also proposes adaptive estimation of the noise level from the blurry features. The experiments show that the proposed method outperforms existing state-of-the art deconvolution methods.

Strengths: Non-blind deconvolution is a still important problem, as it can be used as an important component for solving other problems such as motion deblurring, defocus blur removal, lens blur correction, etc. The paper presents a novel blind deconvolution that outperforms existing state-of-the-art approaches. Deconvolution using deep features is a counterintuitive but interesting idea. Deconvolution using FFTs is possible thanks to the linearity of the blur model. Thus, theoretically, deconvolution using nonlinear deep features does not make sense. On the other hand, the paper shows that it is still possible and improves the deconvolution quality significantly in Table 3. It would also be interesting to see how linear the learned deep features are. Another interesting experiment would be to learn linear features instead of deep nonlinear features and compare their performance against that of deep features. The results shown in the supplementary material are also impressive. Despite the simple structure without any iterations, the paper shows impressive results even for blurry images with severe noise and outliers.

Weaknesses: It seems that the evaluation is not fair. In Sec. 4, the proposed method is compared with various learning-based methods. However, the paper says that only three of them [6,33,47] are retrained using the same training dataset as the proposed one, and it seems that other recent non-blind approaches are not retrained. As the performance of learning-based methods can significantly vary with respect to the training dataset, this must be clearly addressed. The proposed method is limited to the uniform blur model as it is based on the wiener deconvolution.

Correctness: I think so.

Clarity: The paper is well written and easy to understand.

Relation to Prior Work: Yes.

Reproducibility: Yes

Additional Feedback: The paper says the multi-scale feature refinement network shares the parameters across different scales. However, the input feature maps h^1 and h^2 have different numbers of channels. Thus, the subnetworks for l=1 and l=2 should have different numbers of parameters. Then, does the subnetwork for l=1 have additional parameters that are not shared with different subnetworks? == After rebuttal == I am satisfied with the rebuttal, and have no objection for accepting this paper.


Review 4

Summary and Contributions: This paper proposes a non-blind image deblurring method using a learned feature-based Wiener filtering in a neural network framework. Unlike the conventional Wiener deconvolution approach which is performed in the standard image space, the proposed method performs the Wiener deconvolution in the feature space which is learned using neural networks. In addition, they adopt a multi-scale refinement architecture in a coarse-to-fine manner to recover small scales and fine details in images. The paper presents extensive experimental results using blur images with noise, and shows that their method achieves the best performance in non-blind image deblurring among various recent state-of-the-art methods. The main contribution of this paper is that it is the first study to utilize the Wiener deconvolution method for CNN-based non-blind image deblurring.

Strengths: 1) They performed a Wiener deconvolution in the feature space instead of the standard image space, thus it could be more robust to noise as well as inaccurate blur kernels to some extents. To show the robustness of the feature-based Wiener deconvolution against image-based methods, the authors performed some experiments. Table 3 in the original manuscript and Table 8 in the supplementary material show various ablation studies to prove the effectiveness of the feature-based method compared to the original image-based method. When comparing “Wiener_D” and “Ours_G” or “Ours_I+G”, it is clear that the proposed feature-based Wiener deconvolution is more effective in generating more accurate results than image-based methods in terms of PSNR and SSIM. 2) To recover more fine details, they adopt the multi-scale feature refinement module. Table 4 shows the effectiveness of the proposed multi-scale feature extraction module. The proposed method achieved higher values of PSNR and SSIM when using the multi-scale feature refinement module. 3) The authors show that their method is robust to different levels of noise. From Table 1 and Table 2, the proposed method shows the best accuracy for various noise levels among state-of-the-art methods. 4) As shown in Table 5, the authors show the robustness of the proposed method for inaccurate blur kernel estimation. To this end, they separated the kernel for training and testing using ground truth blur kernel and estimated kernel using [41] as well as mix of them. Their method is more accurate about 0.7dB in PSNR than Vasu et al.[41].

Weaknesses: 1) In Table 3, the method "Ours(I+G)" which used hand-crafted features such as intensity and gradient shows PSNR/SSIM values (36.52/0.9583). The results using the proposed feature-based method ("Ours") show the best PSNR/SSIM(36.90/0.9614), the difference between them is 0.4dB. I think this improvement (0.4 dB) is somewhat small and marginal, considering that using the learned-feature for Wiener deconvolution is the main contribution of this paper. 2) Although they show that their method is robust to noise, it still has limitation of the Wiener filter method, in that the accuracy of the Wiener deconvolution is severely dependent on the estimation of the signal-to-noise ratio (SNR). In the supplementary material, the authors explained that the level of the signal (E[|Fx|^2]) and noise (E[|Fn|^2]) in a feature space is estimated using the standard deviation and the mean filtering of Fy. However, I think this simple approach is not so robust to estimate noise level when the noise is not stationary and different to each region in the images.

Correctness: 1) Lines 28-30. The authors claim that “the underlying deep models for image denoising are not specifically optimized for image deblurring”. However, if one sequentially trains conventional denoising models and deblurring models to remove both the noise and blur artifacts respectively, some of the concerns the authors raised could be resolved. 2) It is unclear that the feature extractor in the proposed method plays a role of denoiser. If so, there would not be much noise in the extracted features, and the necessity of using the proposed Wiener deconvolution is weakened. Then, the Wiener deconvolution module could be replaced simply by a sequence of conventional convolutional neural networks. To clarify this simple question, I recommend the authors compare the learned features for blurry images with noise and without noise as was done in Fig. 6 in the supplemental material to see the role of the feature extractor. To be specific, it would be quite interesting to see the learned features in the feature extraction module according to the noise level.

Clarity: Overall, the paper is written well, and easy to understand the main idea. However, there are some unclear points as follows: 1) There is no explanation why the feature-based Wiener deconvolution is better than image-based method. For example, is the proposed feature-based method better for estimating more accurate signal and noise level? 2) In Table 3, the method "Ours w/o Wiener" is not clear. Is the model "Ours w/o Wiener" guided by the input blur kernel? If you guide the kernel, how do you guide it to the network? If not, I think this is not a fair comparative experiment for non-blind deblurring. 3) This method shows robustness to both noise and inaccurate blur kernel estimation. I recommend the authors clearly state why this method outperforms previous method in terms of noise and inaccurate blur kernel. 4) There is no explanation of signal and noise level estimation in the original manuscript, although it is in the supplementary material. In the Wiener filter-based method, I think the SNR estimation process is crucial. Thus, I recommend the authors clearly state the process in the main manuscript. 5) It is unclear how the accuracy of the signal and noise level estimation affects the performance of this method, because the Wiener filtering requires accurate SNR (signal-to-noise ratio) estimation. 6) Because Wiener filter is a linear estimator, I'm wondering if the Wiener filter module can be replaced with a simple conventional CNN modules which could be learned similarly to the Wiener deconvolution. 7) It is unclear that the improvement of the proposed method in Figure 3 is due to multi-scale module or feature-based Wiener deconvolution module. 8) For color test images, the authors tested images with only 1% noise. I recommend the authors include test color images with more noise (up to 5% noise level) as the authors did for the gray-scale color image dataset [19]. 9) In line 190-193, the authors state that only three methods [6, 33, 47] are trained using the same training set as the proposed method. Then, I’m wondering if the other methods (e.g. [47],[50],[36]...) are not trained with the same training set. If the other methods are trained using different training sets, I think the reasonable explanation should be included in the paper.

Relation to Prior Work: Related work section is well-written and clearly structured. In particular, the proposed algorithm is similar to Son and Lee [38] in that it combines the Wiener deconvolution and convolutional neural network. Son and Lee [38] use the Wiener deconvolution as a preprocessing step in the standard image space and then neural network is trained with this deconvolved feature. Different from [38], the submitted paper suggests to use the Wiener deconvolution in feature space. To be specific, the paper’s method extracts the feature first using neural network, and performs the Wiener deconvolution process in the feature space, then the deblurring those deconvolved feature using multi-scale network.

Reproducibility: Yes

Additional Feedback:

[Author Response · NeurIPS 2020]

**(R1, R3) Why it makes sense to deblur the extracted features?** The derivations of the feature-based Wiener
deconvolution strictly hold in a linear feature space. As stated in L136–142, we are interested to examine how to
leverage powerful learned feature extractors $\{\mathbf{F}_i\}$ beyond hand-crafted ones. To this end, we develop a feature extraction
network (FEN) to estimate $\{\mathbf{F}_i\mathbf{y}\}$. As the FEN with ReLUs is piece-wise linear [17, 25], the linearity assumption of
the Wiener filter holds locally. Violations can be successfully compensated by the feature refinement. To evaluate the
feasibility and effectiveness of this piece-wise linear FEN, we have analyzed the effect of using features extracted by the
FEN and hand-crafted features on the classical Wiener deconvolution in Tab. 8 of the supplementary. The results show
that the method with the features extracted by the FEN performs much better than those with hand-crafted features,
increasing the PSNR by at least 0.72dB on the dataset of [19] and 1.11dB on the dataset of [22]. We will clarify this.
**(R1, R3) What if the blur is non-uniform, or the linear model assumption fails?** Although our method is based on a
uniform blur model, it can be extended to handle non-uniform blur by applying the local uniform approximation method
[53].[1] As pointed out in [52], the gamma correction and non-linear CRF can be corrected before the deconvolution
steps. We will discuss these in detail and add corresponding results in the revised paper.
**(R1, R3) Parameters of the feature refinement.** Our feature refinement network shares the parameters across all
scales except for the first encoder block, as the input channel number for scale $l$=1 and $l$>1 differs (Eq. 9). For the other
blocks (except the first), the number of features are the same and thus the parameters can be shared.
**(R1, R4) Marginal improvement of learned features in Tab. 3?** The reason why the improvement in Tab. 3 is not so
prominent is because *Ours* and *Ours*$_{I+G}$ are incorporated with our multi-scale feature refinement, which is good at
recovering clear images from deconvolved features and can compensate some errors caused by hand-crafted features.
However, we have shown that using learned features is much more effective for applying the Wiener deconvolution than
using hand-crafted features, cf. 34.78dB for *Wiener*$_D$ *vs.* 33.37dB for *Wiener*$_{I+G}$ on [19] in Tab. 8 of the suppl.
**(R1) Will the method work at higher noise levels?** Our PSNR results are 25.57 (15%), 24.01 (20%), 21.79 (30%) on
the dataset of [19], which are 0.93dB, 1.33dB, and 0.82dB higher than the best results among the competing methods.
**(R1) Why PSNRs in Tabs. 3 and 5 are so different?** Tabs. 3 and 5 are evaluated on [19] and [16], respectively, with
1% Gaussian noise, where the kernel size ranges from $13 \times 13$ to $27 \times 27$ and $51 \times 51$ to $101 \times 101$. We will clarify.
**(R2) Contribution and motivation.** We develop a simple and effective deep learning-based non-blind image deblurring
method by integrating the domain knowledge of image deconvolution. The contributions are summarized in L50–61.
The motivation to use Wiener deconvolution is that we can use it to derive an effective feature-based deconvolution
operator with SNR estimation, which can be effectively embedded into an end-to-end network for better deblurring.
**(R3, R4) Some recent approaches are not retrained.** As we use the same training dataset as [49], we do not retrain
this model. The training codes of [8, 14, 38, 48] are not available, hence we do not retrain their models. However, we
have finetuned their adjustable parameters for different datasets and noise levels for best results. We will clarify this.
**(R4) Role of feature extractor.** The feature extractor is not a denoiser but is used to provide more useful information
for the Wiener deconvolution (L131–142). We further compare the learned features for blurry images w/o and w/ noise
($1\%, 5\%, 10\%$) and find that the higher the noise level of the blurry image is, the more noise the learned feature contains.
We will discuss in detail and visualize the learned features for various noise levels in the revised paper.
**(R4) Why feature-based Wiener deconvolution is better than image-based method.** First, the feature-based Wiener
deconvolution can utilize more useful information beyond the image intensity to better constrain the deconvolution
process (L110–114). Second, finer-scale detail is better modeled in the feature space. Especially, the deep feature
extractor in our end-to-end network can adaptively learn useful features for the final image restoration (Figs. 3 and 6).
**(R4) Robustness to noise & inaccurate kernels vs. competing methods.** First, benefitting from the proposed feature-
based Wiener deconvolution, our method can adaptively estimate the SNR from blurry features. Second, our end-to-end
network facilitates the feature extractor learning useful features for deconvolution with fewer artifacts and benefits the
feature refinement in handling deconvolved features to reconstruct clearer images. The robustness is also evaluated on
real-world images in Figs. 1, 5, 17–19 (suppl.), where the noise is unknown and estimated kernels are inaccurate.
**(R4) Effect of SNR estimation.** The robustness and effectiveness of the SNR estimation have been demonstrated in
Tabs. 1–2, 9 (suppl.) and Figs. 4–5, 11–19 (suppl.). We further carry out a sensitivity analysis w.r.t. the SNR estimation
using [19] by adding 0–20% perturbation to our estimated SNR (no retraining). The PSNRs differ no more than 0.06dB,
suggesting the robustness of our method to the SNR estimate. In addition, as we do not model spatially-variant noise,
our method may not be so robust to non-stationary noise. This is worth further study.
**(R4) "Ours w/o Wiener" is not clear.** This is the method that removes the Wiener deconvolution in our model, thus
is not guided by the blur kernel. We use this baseline to illustrate the importance of the Wiener deconvolution on the
end-to-end network for non-blind image deblurring, which can effectively incorporate the kernel information.
**(R4) Improvement of the proposed method in Fig. 3.** As both the methods in Fig. 3 use the same multi-scale feature
refinement, the improvement is due to the feature-based Wiener deconvolution module.
**We thank the reviewers for constructive and detailed comments.** We will revise the paper according to the above
responses and add all other suggested references & experimental results and correct the unclear statements as suggested.
[1][53] O. Whyte, J. Sivic, and A. Zisserman. Deblurring shaken and partially saturated images. IJCV, 110(2):185-201, 2014.

[Meta-Review · NeurIPS 2020]

Overall, the reviewers were positive about the paper: the experimental results are very good, and the idea of using Wiener deconvolution in the feature space is interesting. After the rebuttal and discussion, the reviewers unanimously voted for acceptance. Please put the clarifications in the rebuttal into the final version.